# Genetic Variability Assessment of a Diploid Pre-Breeding Asparagus Population Developed Using the Tetraploid Landrace 'Morado de Huétor'

Verónica García, Patricia Castro, Teresa Millán, Juan Gil and Roberto Moreno *

Department of Genetics, University of Cordoba, Campus de Rabanales, C-5, 14071 Cordoba, Spain
* Correspondence: g12mopir@uco.es

**Abstract:** Different studies have reported a narrow genetic base for garden asparagus (*Asparagus officinalis* L.) due to its common origin, a diploid population ('Purple Dutch'). The present study focused on the development of new diploid plant material that may be useful to widen the genetic base of the crop by using a tetraploid landrace 'Morado de Huétor' (*A. officinalis* × *A. maritimus*). With this purpose, a diploid pre-breeding population (*n* = 1000) carrying introgressions of 'Morado de Huétor' has been obtained. This new population derived from crosses under open pollination of a parental collection (*n* = 77) that was developed in a previous study. The parental collection derived from the first backcrossing using different diploid cultivated plants as a recurrent parent and 'Morado de Huétor' as a donor. The genetic diversity of the pre-breeding population was assessed using a set of EST-SSR markers (AG7, AG8, TC1, TC3, TC7, TC9) in a collection of plants (*n* = 57), which was randomly sampled in the pre-breeding population. The results were compared to previous data obtained from the parental collection, a set of current diploid asparagus cultivars and the landrace 'Morado de Huétor'. The average of PICm (Polymorphic Information Content) values obtained in the pre-breeding population (0.75) resulted higher than the value obtained in the diploid cultivars (0.63) but lower than in 'Morado de Huétor' (0.83). Twenty-two alleles (52.4%) detected in the new diploid population were specific from 'Morado de Huétor'. Principal Coordinate Analyses (PCoA) revealed that the new population had a genetic diversity distribution different from the current cultivars. This new population was also evaluated for different morpho-agronomic traits (earliness, stalk number, branching height and stalk thickness) for two years. Significant differences among plants (*p* < 0.001) were found for these five traits and, therefore, a genotype variation is suggested. As a result, 71 plants were selected to develop a breeding base population. The genetic variability of those selected plants was also analyzed and similar genetic variability to the pre-breeding population was obtained. The results obtained in this study show that this new population could be used to enlarge the genetic base of the current diploid asparagus cultivars.

**Keywords:** Asparagus improvement; pre-breeding; breeding base; genetic resources; landrace; genetic variability; EST-SSR markers

## 1. Introduction

Garden asparagus (*Asparagus officinalis* L. 2n = 2x = 20) is an important horticultural crop worldwide, cultivated mainly in regions with a temperate or subtropical climate such as Asia, Europe and the American continents. The cultivated area of this species is similar to other important vegetable crops such as garlic, green bean or eggplant [1]. *A. officinalis* is the most economically important species of the *Asparagus* genus and the only one cultivated as vegetable [2]. Additionally, the young spears of some wild *Asparagus* spp., such as *A. acutifolius* L., *A. albus* L., *A. verticillatus* L., *A. acerosus* Thunb. ex Schult. & Schult. f. or *A. laricinus* Burch, among others, are collected for self-consumption or sold in local markets [3].

Several *Asparagus* spp. including *A. officinalis* have been used as medicinal plants since ancient times, even prior to their use as a vegetable [4,5]. Some pharmacological properties have been well documented in the cultivated species, highlighting the antioxidant, anticancer or diuretic effects [6,7]. According to the beneficial properties for health proposed for this species, it has been listed as a nutraceutical food [7].

Most current cultivars are diploid and derived from a Dutch population ('Violet Duch') developed in the 18th century [8,9]. This could explain the reduced genetic base of the current cultivars described in several studies employing different molecular markers such as isozymes [8,10,11] and DNA-derived markers such as RAPDs [12,13] or SNPs [14].

Several stresses cause economic losses in this crop including viruses and fungal diseases, mainly: crown and root rot, purple spot, rust and phytophthora root rot. Other abiotic stresses affecting this crop are salinity, drought or acidic soil. Tolerances or resistances to these stresses have been found in some wild *Asparagus* spp. such as *A. maritimus* (L.) Mill., *A. acutifolius*, *A. prostratus* Dumort, or *A. verticillatus*, among others [15–17]. Additionally, crop wild relative species could be a source of genes controlling bioactive compound profiles absent in current cultivars [18,19]. In this sense, there is currently an increasing demand for health-promoting foods that may reduce the risk of some diseases. Accordingly, these genetic resources may be useful in asparagus breeding for developing new cultivars that could respond to the present and future demands from farmers and consumers.

The use of wild relative species or landraces in the development of new asparagus cultivars has lagged behind other vegetable crops with similar economic importance. In this sense, new improved materials have been developed from landraces in green bean [20] or eggplant [21] through backcrossing or classical selection. Nowadays, the problems that might happen when these traditional plant breeding methods are employed, such as linkage drag, can be overcome by the biotechnological approaches available today [22] and others that will surely be devised in the future [23].

The studies aimed at the development of new germplasm suitable for the genetic improvement of this crop have been scarce so far (see review by Moreno et al. [24]). With this aim, Riccardi et al. [25], using the in vitro anther culture technique, developed new diploid (di-haploid) germplasm carrying introgressions from two wild tetraploid populations (*A. acutifolius*, *A. maritimus*) and a tetraploid landrace ('Violetto d'Albenga'). Additionally, diploid plants carrying germplasm from the tetraploid landrace 'Morado de Huétor' were obtained by backcrossing using the diploid cultivars as a recurrent parent by Castro et al. [26] and using in vitro anther culture by Regalado et al. [27]. More recently, Plath et al. [28] detected AV-1 resistant plants in the progenies resulting from the first backcrosses with different interspecific hybrids, which were obtained between *A. officinalis* and wild related species (*A. maritimus* (6x), *A. pseudoscaber* (6x), *A. prostratus* (4x)). The most comprehensive study aimed at developing new germplasm has been recently published [29]. These authors developed a hexaploid pre-breeding population carrying introgressions from a set of wild polyploid-related species (*A. maritimus*, *A. pseudoscaber* Grec., *A. brachyphyllus* Turz., *A. machrorrizus* Pedrol & al.) into 'Morado de Huétor', a land race obtained from a natural cross between garden asparagus (*A. officinalis*) and a wild relative species (*A. maritimus*) [30].

'Morado de Huétor' is a landrace cultivated in Granada (South of Spain) along the Genil river valley. This geographical area is characterized by its temperate climate with hot and dry summers. A high variability for important agronomic traits (earliness, yield, tolerances to different stresses) or morphological traits of interest in this crop, such as spear morphology, color, tastes or branching height, have been found in the farmers' field where this landrace is cultivated [13,31]. In addition, higher profiles of bioactive compounds such as saponins and flavonoids than that obtained in some commercial hybrids have been detected in this landrace [32,33].

Considering all this information, the tetraploid landrace 'Morado de Huétor' could be a valuable genetic resource for the genetic improvement of this crop. With this aim, first, our research group explored the possibility of developing triploid hybrids between plants

of the tetraploid 'Morado de Huétor' and plants of diploid cultivars [34]. The first result of using this landrace in the development of new asparagus cultivars was an octoploid clonal hybrid ('HT801') that derived from a cross between two octoploid plants (female and male) found in two different farmers' plots located in the area where this landrace is cultivated [35]. Differences for several important morpho-agronomic traits in this crop and higher bioactive compound profiles than in current cultivars have been described for this octoploid cultivar [35,36]. This study focused on the development of a pre-breeding diploid population carrying germplasm from the tetraploid 'Morado de Huétor'. The genetic diversity of this population was also assessed employing a set of EST-SSR markers.

## 2. Materials and Methods

### 2.1. Plant Material

An experimental field trial plot established with 77 diploid plants with germplasm of the tetraploid landrace 'Morado de Huétor' was employed in the development of a diploid population ($n$ = 1000) (Figure 1). The parental collection ($n$ = 77) of this new population was the result of a first backcrossing using diploid current cultivars ('Atlas', 'Grande', UC-157' 'Mary Washington') as a recurrent male parent [26]. This parental collection was growing under open pollination conditions during the 2015 vegetative season. Seeds were picked up from all female plants ($n$ = 31) in autumn. A similar number of seeds of these female plants was sown in a seedbed placed under greenhouse conditions. These seeds represent the second generation from the first backcrossing previously mentioned. In the spring of 2016, a balanced number of seedlings ($n$ = 1000) was transplanted (0.5 m $\times$ 1.5 m) in an experimental field located in a Research Center ('Alameda del Obispo') belonging to the Regional Government of Andalusia (IFAPA), Córdoba (Spain).

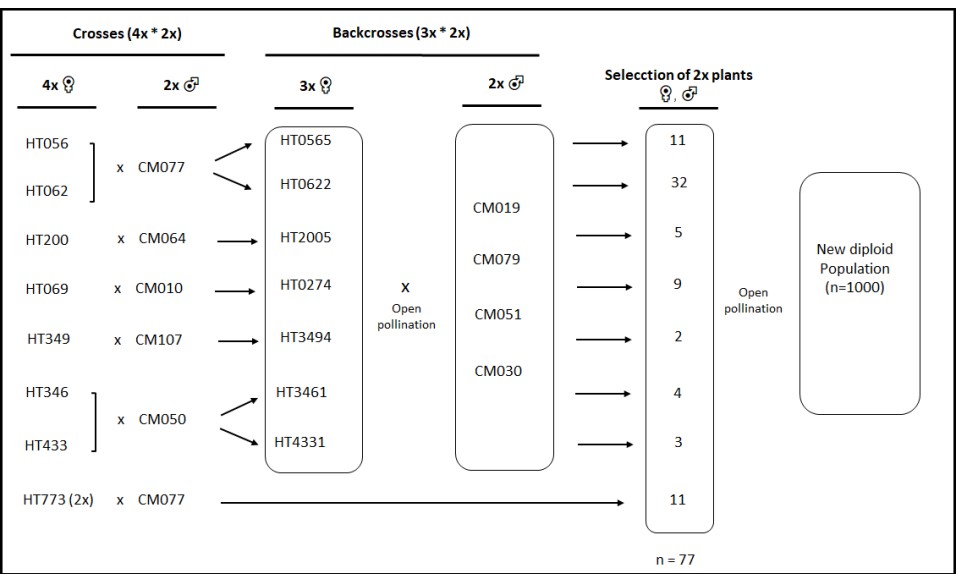

**Figure 1.** Scheme of crosses and backcrosses carried out to develop the diploid pre-breeding population evaluated in this study. Seven tetraploid plants [HT056-HT433], one diploid (HT773) from the tetraploid landrace 'Morado de Huétor' and 9 diploid plants from different diploid cultivars were employed in the crosses: CM010 ('Baitoru'), CM019 ('Atlas'), CM030 ('M. Washington'), CM050 ('Jaleo'), CM051 ('UC-157'), CM064 ('J. Marionnet 2001'), CM077, CM079 ('Grande') and CM107 ('Haideru').

### 2.2. SSR Marker Analysis

SSR marker analysis was carried out according to Garcia et al. [29]. Briefly, the assessment of the genetic variability of the population ($n$ = 1000) obtained in this study was estimated using a sample of 57 individuals that were randomly chosen from the new population. The total genomic DNA of each individual plant was isolated using a modified CTAB method [37]. Six EST-SSR markers (AG7, AG8, TC1, TC3, TC7 and

TC9) previously developed by Caruso et al. [38] were employed in the genetic variability study. These six markers were selected among a set of 12 markers because of their high polymorphic information content (PIC) and clear and reproducible peak pattern in all individuals. In previous studies, they were useful for analyzing asparagus cultivars as well as the landrace 'Morado de Huétor' [26,39,40]. The PCR products were separated using an automated capillary sequencer (ABI 3130 Genetic Analyzer; Applied Biosystems/HITACHI, Madrid, Spain). The results of the sample of 57 individuals obtained from the new diploid population were compared with the parental collection (*n* = 77), the landrace 'Morado de Huétor' (*n* = 38) and 63 diploid plants from different cultivars (around eleven plants per cultivar). 'Atlas', 'Grande', 'UC157', 'Plaverd', 'Steline' and 'Thielim' were the cultivars used [29,41]. The alleles were scored as presence (1) or absence (0) and a binary data matrix was created. The PIC (Polymorphic Index Content) marker values were calculated employing the formula PICm = $1 - \Sigma \, Pj^2$, where *Pj* is the band frequency of the *j*th allele. A Principal Coordinate Analyses (PCoA) was carried out using GENALEX 6.5 software [42]. To estimate the genetic variation observed in the biplot from PCoA, the mean value of pairwise distance was calculated in each population. The smaller the mean distance, the smaller variation and vice versa. Student's t-test was also employed to compare the mean values.

### 2.3. Assessment of Morpho-Agronomic Traits

The diploid population (*n* = 1000) was phenotyped for five morpho-agronomic traits for two years. The seedbed was sown in autumn 2015 and the field trial was stablished in early spring 2016. The traits and years of the evaluations are presented in Table 4. The evaluations were performed following the methodology described by Garcia et al. [29]. Briefly, earliness of each plant was assessed in a single day at the beginning of spring by both the spear production and the phenological stage reached by the spears. In autumn, the stalk number per plant, branching height and stalk thickness were assessed. An analysis of variance was carried out for the two years according to the model ($xij = \mu + Ai + Yj + eij$) where *xij* is the individual date, *μ* the general mean, *Ai* the effect of *i*th plant, *Yj* the effect of *j*th year and *eij* is the residual error.

According to the results of the evaluations, 71 plants (50% female) were selected because of their superior morpho-agronomic performance. All these selected plants were crossed under open pollination with the aim of developing a breeding base population in the future. For that purpose, seeds from all female plants (*n* = 35) were harvested at the end of the vegetative season. A total of 67 of the 71 plants were analyzed using the set of the EST-SSR markers previously mentioned. The genetic diversity of the selected population was compared with the samples taken at random (*n* = 57) from the pre-breeding population developed in this study.

## 3. Results

### 3.1. Genetic Diversity Analysis

The results of the genetic variability study of the new diploid population employing the set of EST-SSR markers is shown in Table 1. All six SSR markers resulted polymorphic and informative (PICm > 0.5). In the new diploid population, the number of alleles per marker varied from 4 to 11 for AG7 and TC3, respectively. Mean PICm was 0.75, with 0.61 (AG7) and 0.85 (TC3) as the lowest and highest values, respectively. Twenty-two alleles (52.4%) found in the diploid population (*n* = 57) were specific from 'Morado de Huétor', confirming the presence of germplasm of this landrace in this population.

**Table 1.** Results of the genetic variability study. Polymorphic information content (PICm) and number of alleles detected in 'Morado de Huétor', asparagus cultivars, the parental population and the new population obtained in this study.

| EST-SSR Locus | 'Morado de Huétor' (*n* = 38) | | Diploid Cultivars (*n* = 63) | | Parental Popul. (*n* = 77) | | New Diploid Popul. (*n* = 57) | |
|---|---|---|---|---|---|---|---|---|
| | Nº. Alleles | PICm | Nº. Alleles | PICm | Nº. Alleles | PICm | Nº. Alleles | PICm |
| AG7 | 8 | 0.77 | 3 | 0.52 | 4 | 0.64 | 4 | 0.61 |
| AG8 | 14 | 0.88 | 4 | 0.49 | 7 | 0.68 | 7 | 0.73 |
| TC1 | 14 | 0.87 | 3 | 0.61 | 6 | 0.66 | 6 | 0.73 |
| TC3 | 15 | 0.84 | 5 | 0.74 | 12 | 0.78 | 11 | 0.85 |
| TC7 | 8 | 0.81 | 6 | 0.72 | 7 | 0.71 | 7 | 0.75 |
| TC9 | 13 | 0.84 | 4 | 0.62 | 7 | 0.83 | 7 | 0.83 |
| Total | 72 | | 25 | | 43 | | 42 | |
| Mean | **12.0** | **0.83** | **4.2** | **0.61** | **7.2** | **0.72** | **7.0** | **0.75** |

Comparisons among the populations evaluated in this study pointed out that the total number of alleles detected in the new population (42 alleles) was higher than those detected in the current diploid cultivars (25 alleles) and lower than the tetraploid landrace 'Morado de Huétor' (72 alleles). Additionally, the average PICm value of the new population (0.75) was higher than the value observed in the diploid cultivars (0.61) but lower than 'Morado de Huétor' (0.83) (Table 1). According to these results, the new diploid population displayed higher allelic diversity than the current cultivars. Principal Coordinates Analyses (PCoA) were applied to evaluate the genetic variability among the different genetic stocks: the tetraploid landrace 'Morado de Huétor', the diploid cultivars, the parental population and the new diploid population (Figure 2). Based on the PCoA analysis, 58.7% of the total variation is accounted for by the two main principal coordinates. The results of the PCoA analysis showed two well-defined groups: the diploid populations and the tetraploid landrace 'Morado de Huétor'. Additionally, differences in the pattern of the genetic variability among diploid populations were observed in the biplot. The set of commercial cultivars showed less dispersion than the new diploid population developed in this study and the parental population. These results reveal a greater genetic variability for the new diploid population than for the diploid commercial cultivars, similar to the parental population. To estimate the genetic diversity observed in the biplot, the mean genetic distance between pairs of individuals within each population was estimated. A Student's t-test was performed to assess the differences between the mean of the new population and the other ones included in the present study. As a result, the mean distance was significantly higher ($p < 0.001$) in the new diploid population than in the current diploids, 'Morado de Huétor' and original parental populations (Table 2).

**Table 2.** Mean of genetic distances between pairs of individuals obtained in the asparagus populations included in this study.

| Population | N [1] | Mean ± SE [2] | |
|---|---|---|---|
| **New diploid** | 57 | 12.38 ± 0.072 | |
| Parental diploid | 77 | 10.93 ± 0.057 | *** |
| Morado de Huétor | 38 | 18.78 ± 0.157 | *** |
| Diploid cultivars | 63 | 8.31 ± 0.070 | *** |
| Diploid selected plants | 67 | 12.12 ± 0.061 | ** |

**, *** Significantly different $p < 0.01$ and $p < 0.001$ respectively from the new diploid population (bold) by *t*-Student test. [1] Number of individuals per population. [2] Standard error.

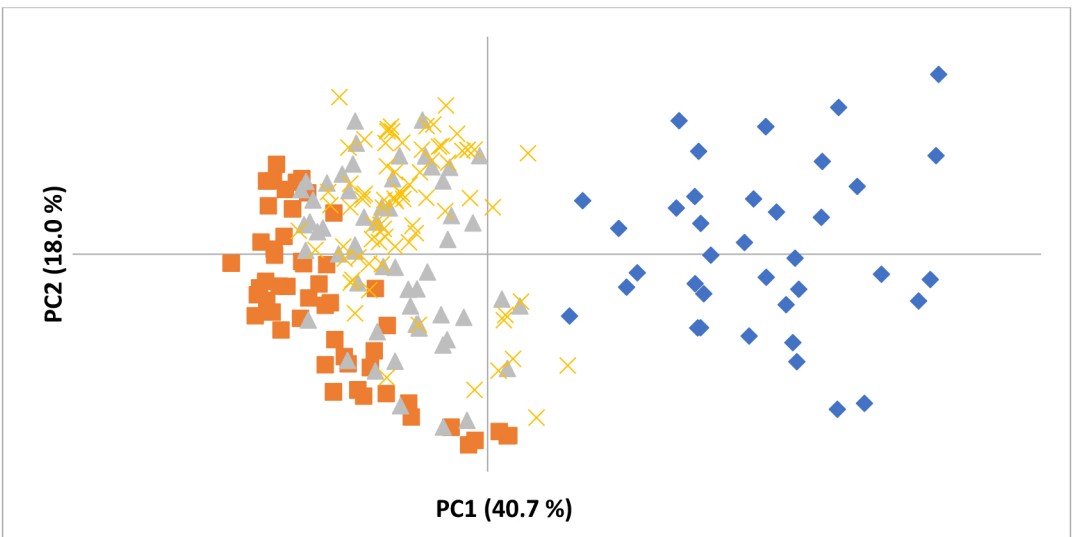

**Figure 2.** Biplot resulted from the PCoA analysis carried out in the tetraploid landrace (♦), diploid cultivars (■), diploid parental population (X) and the new diploid population (▲).

*3.2. Phenotypic Evaluation*

Five morpho-agronomic traits were evaluated in the new diploid population for two years. These traits displayed significant variation ($p < 0.0001$) between individuals. Accordingly, a genotypic variation for these five traits is suggested in this population (Table 3). As a result of the evaluations for the morpho-agronomic traits, 71 plants were selected in the new diploid population. These plants were allowed to flower in the experimental trial conducted under open pollination conditions. The seeds picked up from all female parents ($n = 35$) of these selected plants could be used to develop a breeding base population. The collection of selected parent plants displayed mean values higher than the mean of the total population ($n = 1000$) (Table 4). A total of 67 out of these 71 selected plants were also genotyped with the set of six EST-SSR markers. The genetic diversity of the selected plants was also compared to the whole population ($n = 1000$). PCoA results showed a similar genetic diversity between these two sets of samples selected in the new diploid population (random sampling and selected) (Figure 3). The mean distances were similar, 12.38 in the new population and 12.12 in the selected plants, although significant differences ($p < 0.01$) between them were found (Table 2). This result may be explained by the low mean standard error due to the high number of pairwise distances within each sample.

**Table 3.** Results of analysis of variance for two years for the morpho-agronomic traits evaluated in the new diploid population of cultivated asparagus carrying introgressions from the landrace 'Morado de Huétor'.

| Traits [b] | Mean Square [a] | | |
|---|---|---|---|
| | Year df = 1 | Individuals df = 766 | Error df = 766 |
| Spear number | 471.52 (<0.0001) | 27.71 (<0.0001) | 11.55 |
| Phenological stage | 27.79 (<0.0001) | 1.97 (<0.0001) | 1.37 |
| Stalk number | 18.23 (<0.0001) | 0.22 (<0.0001) | 0.11 |
| Branching height | 3.61 (<0.0001) | 0.10 (<0.0001) | 0.06 |
| Stalk thickness | 1.25 (0.0016) | 0.20 (<0.0001) | 0.06 |

[a] In brackets *p* value. [b] Traits evaluated according to Garcia et al. [29].

**Table 4.** Mean and standard deviation (SD) the five morpho-agronomic traits evaluated in a diploid population of cultivated asparagus with introgressions from the landrace 'Morado de Huétor' (*A. officinalis* × *A. maritimus*). Data from a set of selected plants (*n* = 71) are also presented.

| Trait | Population | Year | N | Mean ± SE | SD |
|---|---|---|---|---|---|
| *Spring* | | | | | |
| Spear number | Total | 2017 | 841 | 5.4 ± 0.13 | 3.81 |
| | | 2018 | 823 | 4.4 ± 0.17 | 4.80 |
| | Selected | 2017 | 71 | 8.1 ± 0.60 | 5.06 |
| | | 2018 | 71 | 8.3 ± 0.80 | 6.72 |
| Phenological stage | Total | 2017 | 791 | 2.3 ± 0.04 | 1.12 |
| | | 2018 | 793 | 2.0 ± 0.05 | 1.43 |
| | Selected | 2017 | 65 | 3.0 ± 0.11 | 0.87 |
| | | 2018 | 71 | 3.0 ± 0.11 | 0.94 |
| *Autumn* | | | | | |
| Stalk number | Total | 2016 | 846 | 1.7 ± 0.01 | 0.39 |
| | | 2017 | 830 | 1.5 ± 0.01 | 0.38 |
| | Selected | 2016 | 71 | 1.8 ± 0.04 | 0.36 |
| | | 2017 | 71 | 1.8 ± 0.05 | 0.44 |
| Branching height | Total | 2016 | 846 | 1.5 ± 0.01 | 0.32 |
| | | 2017 | 829 | 1.4 ± 0.01 | 0.22 |
| | Selected | 2016 | 71 | 1.8 ± 0.04 | 0.34 |
| | | 2017 | 71 | 1.6 ± 0.02 | 0.20 |
| Stalk thickness | Total | 2016 | 846 | 1.5 ± 0.01 | 0.39 |
| | | 2017 | 829 | 1.5 ± 0.01 | 0.31 |
| | Selected | 2016 | 71 | 1.9 ± 0.05 | 0.40 |
| | | 2017 | 71 | 1.9 ± 0.03 | 0.29 |

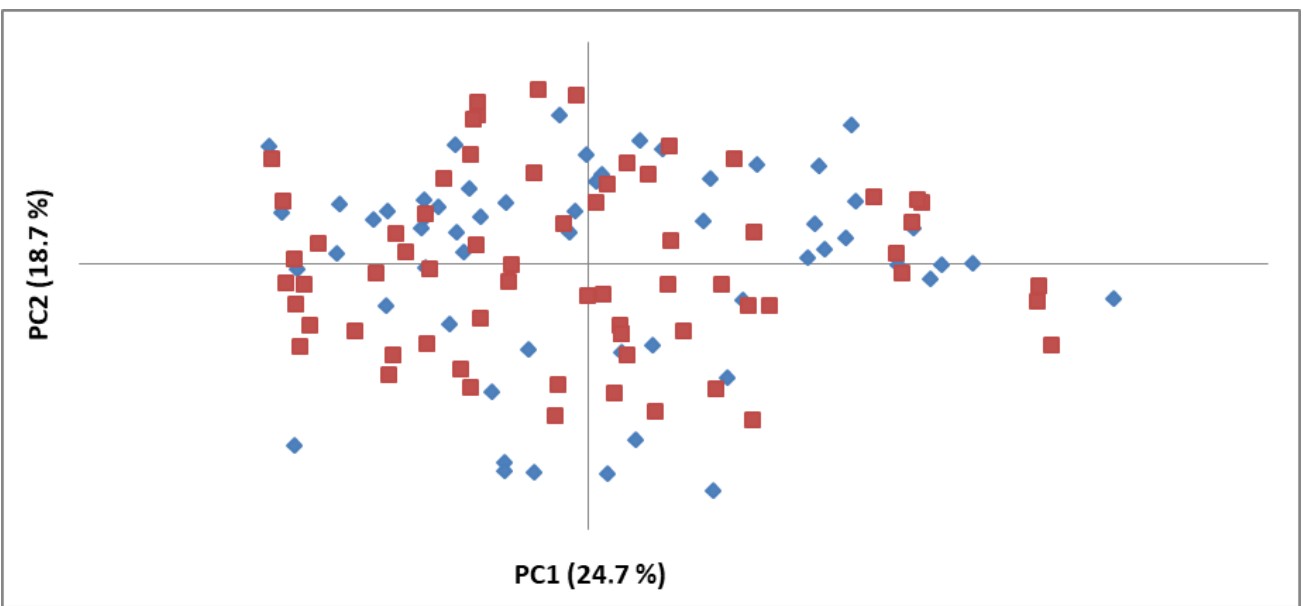

**Figure 3.** Results of PCoA analysis carried out on two different sets of samples selected in the new diploid population: random sampling (◆) superior morpho-agronomic performance (■).

## 4. Discussion

Different studies have highlighted the reduced genetic variability of the asparagus crop so far [8,10–14]. This hypothesis may be explained by the common genetic origin proposed for most current diploid cultivars [8,9]. Given this scenario, there is a clear need to increase the gene pool of this crop that may facilitate its adaptation to the new scenarios gradually imposed by the unquestionable global climate change. To date, plant genetic resources have had an important role in different plant breeding programs of different crops

because they have allowed the development of cultivars with tolerance or resistance to important biotic/abiotic stresses that may affect the crop or have new agro-morphological traits [43,44].

Despite the importance that the genetic resources may have in asparagus breeding, their use with this aim is far behind other vegetable crops [45]. This fact may be in part explained by the difficulty and/or time-consuming nature of working with polyploid genetic resources as a donor in plant breeding focused on the development of diploid cultivars. A huge number of genetic resources of this crop, including wild related species, landraces and wild populations of the cultivated species, are polyploids [16,41,46]. The tetraploid 'Morado de Huétor' is a genetic resource that may have an important role in the genetic improvement of garden asparagus due to its high genetic variability [8,13,32,36,41,47]. Using nuclear ribosomal DNA internal transcribed spacers' polymorphisms, a natural interspecific origin for this landrace from crosses between *A. officinalis* and *A. maritimus* has been proposed [30]. The first attempt of using this landrace as a genetic resource was carried out by our group [34,48]. Our results showed heterosis in experimental triploid hybrids from crosses between plants of this landrace (as female) and plants of current diploid cultivars (4x * 2x). Additionally, to introgress the 'Morado de Huétor' germplasm into diploid cultivars, triploid plants were backcrossed with diploid plants and diploid individuals were selected from their progenies by flow cytometry, and a collection of 77 diploid plants was obtained [26].

In this study, the progeny of that collection (*n* = 77) obtained under open pollination has been used to develop a diploid pre-breeding population carrying introgressions of 'Morado de Huétor'. This new population has been genotyped by a set of EST-SSR markers and around 50% of the alleles detected are specific the landrace 'Morado de Huétor'. Similar results were obtained in the parental collection of the new diploid population [26]. Despite this fact, the new population showed higher genetic variation than the parental collection. The greatest genetic variation detected in the new population compared to parental population could be explained by the new genetic combinations that occurred after the crosses among plants of the origin population. Our results confirm that the tetraploid 'Morado de Huétor' can be used to widen the genetic base of current diploid cultivars. Taking into account the proposed interspecific origin for this landrace, *A. officinalis* × *A. maritimus*, it can be considered that the new diploid population developed in this study carries introgressions from a wild species. Resistance to biotic (virus AV-1, rust) and abiotic stresses (salinity) has been found in *A. maritimus* [15,49,50]. Therefore, the new diploid population obtained in this study could be a good genetic resource for searching genes that may provide resistance for these important stresses in garden asparagus.

On the other hand, there is currently a growing demand for health-promoting foods that may reduce the risk of some diseases. In this sense, studies conducted with 'Morado de Huétor' have shown higher profiles of bioactive compounds in this landrace than in some diploid commercial hybrids [32,33,36]. Jaramillo et al. [51] reported an anti-carcinogenic effect against human colon cancer cells employing steroidal saponins isolates from edible spears of the landrace 'Morado de Huétor'. Accordingly, this breeding population may be useful in the development of new cultivars with different agro-morphological traits and/or higher bioactive compounds profiles.

As a result of the evaluations carried out in the new population for different morpho-agronomic traits, a set of plants was selected based on their superior morpho-agronomic performance. The genetic variability of this set of plants employing EST-SSR markers was similar to the pre-breeding population obtained in this study. Crosses under open pollination conditions were carried out among these selected plants. The seeds from all selected female plants were harvested to generate a diploid breeding base population in the future.

## 5. Conclusions

The diploid breeding population obtained in the present study may offer the opportunity to obtain new parents to be used in the genetic improvement of this crop, which could be helpful to respond to the new challenges of climate change, as well as to the growing consumer demand for health-promoting foods.

**Author Contributions:** Conceptualization, J.G. and R.M.; Formal analysis, V.G. and R.M.; Funding acquisition, T.M. and J.G.; Investigation, V.G., P.C. and R.M.; Project administration, J.G.; Supervision, J.G.; Validation, J.G. and R.M.; Visualization, V.G., P.C. and R.M.; Writing—original draft, V.G.; Writing—review and editing, P.C., T.M., J.G. and R.M. All authors have read and agreed to the published version of the manuscript.

**Funding:** This work has been supported by the Spanish projects AGL2014–57575-R and PID2019-106991RB-100 financed in part by EU funds (ERDF).

**Institutional Review Board Statement:** Not applicable.

**Informed Consent Statement:** Not applicable.

**Conflicts of Interest:** The authors declare no conflict of interest.

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
