# Peer review of "Genetic Variability Assessment of a Diploid Pre-Breeding Asparagus Population Developed Using the Tetraploid Landrace ‘Morado de Huétor’"

_horticulturae, doi:10.3390/horticulturae8100859_

Round 1

Reviewer 1 Report

The work describes the molecular and phenotypic characterization of selected (probably) diploid pre-breeding asparagus populations that emerged from the tetraploid MH. The data presented are based on a long-term program of the research team in Cordoba, some of which has already been published in other journals in a different context and with a different focus.

In many places reference is made to the published preparatory work.

Nevertheless, the present manuscript should also go into more detail, e.g. in the preparation of the material.

In particular, the crossed varieties are important for the reader.

All results were discussed under the hypothesis that the selected plants are diploid (2n=2x=20).

  Even if the described backcross strategy (4x x 2x à 3x; 3x x 2x à 2x) can theoretically be expected to find diploid offspring, a reliable differentiation of diploid and aneuploid plants (e.g. monosomic or disomic additions) using flow cytometry is not secure possible.

In addition, the genomic status of MH has not been finally elucidated. Is MH i) allotetraploid with both genomes A.o.+ A.m. or ii) an autotetraploid A.o. with a unknown part of introgression from A.m.?  

In the first case, there is a risk of losing all the A.m. chromosomes in the back-crossing program with diploid A.o.

In the second case a rare pairing between homeologous chromosomes is possible which can lead to stable introgressions.

The field op-strategy makes tracking more difficult, but increases the chance of finding interesting recombinants due to the mass of crossings.

This all needs to be discussed critically and a final cytological examination of chromosomes in mitosis and meiosis should be planned as definitive evidence.

The value of putative introgression line for asparagus breeding is undisputed, so I consider the manuscript worthy of being published in Horticulturae after a minor revision.

Style:

The authors alternate between the terms "variety(es)" and "cultivar(s)" (L 91,92,97....). This should be applied consistently. Since certified asparagus cultivars (beside the landrace MH) were used in this work, only the term cultivar should be used.   The reviewer is aware that both terms, stalk and spear, are used in public media, laboratories and the asparagus community. Both shoots are botanical. In scientific papers, botanically correct terms should be used.

Abstract:

L25/L160 – privative ??  MH specific?/A.m. specific / non-A.o. allel

Keywords:

 L35 - Breeding base

Introduction:

L 74 – a number of international groups (Italy, Japan, Germany, China..) are working intensively on new germplasm and increasing the genetic base of asparagus

Material and Methods:

L 106 …current cultivars..- how many cultivars? And the names should given

L 107 … op-conditions… -  how was outcrossing of other cultivars or wild relatives prevented

                                               (see also L 147)

L 120…EST-SSR..Why only six ESTs were used? (a very small base to cover all chromosomes)

L 127… set of current cultivars… - how much cultivars, how much plants per cultivar???

                                                            (Ref. 23  4cvs?)

L 138 …two years… how old were the plants during first phenotyping?

L 143   date

Results:

L 162  how much cultivars were used (n=63?) – 63 cultivars?

L 195   Tab.3 -  how you explain the increasing mean distance from the parental to „new diploid“ and

                          „Dipl. selected“?

L 230  Tab.5 – why no control (MH, cultivars) were evaluated ?

Discussion:

-          See general comments

References:

Cited references are complete in text and reference list.

Author Response

Comments and Suggestions for Authors

The work describes the molecular and phenotypic characterization of selected (probably) diploid pre-breeding asparagus populations that emerged from the tetraploid MH. The data presented are based on a long-term program of the research team in Cordoba, some of which has already been published in other journals in a different context and with a different focus.

Nevertheless, the present manuscript should also go into more detail, e.g. in the preparation of the material.

In particular, the crossed varieties are important for the reader.

 Response:  In order to include more details in the preparation of the plant material we have included a new Figure (Fig. 1) in the Material and Methods section. The cultivars employed in all crosses and backcrosses carried out to  develop the plant material used in this study is described in the legend of that figure.

Even if the described backcross strategy (4x x 2x à 3x; 3x x 2x à 2x) can theoretically be expected to find diploid offspring, a reliable differentiation of diploid and aneuploid plants (e.g. monosomic or disomic additions) using flow cytometry is not secure possible.

Response: We agree that a reliable differentiation of diploid and aneuploid plants (e.g. monosomic or disomic additions) using flow cytometry is not secure possible, and aneuploid plants can be present in the population. However, in previous work (Castro et al, 2014) we considered and discussed about that fact (see below). Please, keep in mind that the current work is a continuation of the work presented in Castro et al 2014.

It is assumed that gametes carrying an extra-long chromosome are genetically more unbalanced than gametes with a shorter chromosome and results in increased gametic and zygotic lethality (Singh, 2010). The transmission of extra chromosomes in plants is considered different in female and in pollen. Theoretically, about 50% of the trisomic plants are expected in the progeny from 2x + 1 × 2x, but this percentage is rarely observed (Singh, 2010). The transmission of extra chromosomes through pollen is usually very low. Pollen with n + 1 chromosome constitution is unbalanced and generally unable to compete in fertilization with pollen carrying the balanced, n, chromosome number (Gupta, 2007; Singh, 2010). According to Löptien (1979), in asparagus, the transmission rates of an extra chromosome seems to be higher for the eggs than for pollen and the larger chromosomes have lower pollen transmission than the smaller ones. In a practical point of view, collecting seeds from female plants of the diploid population would favor the disappearance of the trisomic plants after several generations, resulting in a completely diploid population. In spite of the lack of accuracy of flow cytometry to distinguish between diploid and trisomic asparagus in this study, this method is more user-friendly and faster than microscope observations.”  

In addition, the genomic status of MH has not been finally elucidated. Is MH i) allotetraploid with both genomes A.o.+ A.m. or ii) an autotetraploid A.o. with a unknown part of introgression from A.m.?  

In the first case, there is a risk of losing all the A.m. chromosomes in the back-crossing program with diploid A.o.

In the second case a rare pairing between homeologous chromosomes is possible which can lead to stable introgressions.

The field op-strategy makes tracking more difficult, but increases the chance of finding interesting recombinants due to the mass of crossings.

This all needs to be discussed critically and a final cytological examination of chromosomes in mitosis and meiosis should be planned as definitive evidence.

Response: In a previous work we considered the origin of tetraploid landrace MH as a natural hybridization between A. officinalis (2x) and A. maritimus (6x) that has evolved by farmers’ hands from its natural origin to the present days (Moreno et al, 2008). A. maritimus is a species closely related to the crop. In fact, some authors mentioned it as subspecies of A. officinalis. We have named this species together with other wild species (A. prostratus (4x), A. brachiphylus (6x), ...) as the “officinalis Group” (Castro et al, 2013). Polyploidy is frequent in the Asparagus genus probably due to the formation of unreduced gametes (Camadro 1992; Regalado et al 2015). In a previous study (Castro et al 2013) crosses between an hexaploidy plant of MH and A. maritimus were made, and high pollen fertility and regular meiosis behavior were observed in those plants (Castro et al, 2013). These results suggest that MH is an autotetraploid. Different studies employing DNA molecular markers (Moreno et al 2008; Castro et al 2013) and characterization of the saponin profile (bioactive compound) (Jaramillo-Carmona et al 2017) support the hypothesis that the landrace ‘Morado de Huétor’ carries introgressions from A. maritimus. 

The value of putative introgression line for asparagus breeding is undisputed, so I consider the manuscript worthy of being published in Horticulturae after a minor revision.

Style:

The authors alternate between the terms "variety(es)" and "cultivar(s)" (L 91,92,97....). This should be applied consistently. Since certified asparagus cultivars (beside the landrace MH) were used in this work, only the term cultivar should be used.   The reviewer is aware that both terms, stalk and spear, are used in public media, laboratories and the asparagus community. Both shoots are botanical. In scientific papers, botanically correct terms should be used.

 Response: Agreed. It has been modified in the manuscript.

Abstract:

L25/L160 – privative ??  MH specific?/A.m. specific / non-A.o. allel

Response: It has been modified in the manuscript.

Keywords:

 L35 - Breeding base

Response: It has been modified in the manuscript.

Introduction:

L 74 – a number of international groups (Italy, Japan, Germany, China..) are working intensively on new germplasm and increasing the genetic base of asparagus

Response: We thanks the reviewer for the suggestion. We have included in the manuscript a research work focused on the development of new germplasm, which was performed by a German group. Also, it is cited in the text a research work conducted by an Italian group. Other authors from Japan have obtained interspecific hybrids with wild species but, as far as we know, no subsequent studies focused on the development of new germplasm employing these interspecific hybrids have been published so far. Several works from researchers from China were focused on evaluating new varieties and developed new varieties but there is no record of the use of new germplasm in the development of those varieties.

 Material and Methods:

L 106 …current cultivars..- how many cultivars? And the names should given

Response: The information requested has been included in Material and Methods section.

107 … op-conditions… -  how was outcrossing of other cultivars or wild relatives prevented

                                               (see also L 147)

Response: In order to prevent outcrossing of other cultivars or wild relatives, all the plants were planted in an isolated macro tunnel cover with a netting. Pollinators were introduced in the macro tunnel to favor the open pollination.

L 120…EST-SSR..Why only six ESTs were used? (a very small base to cover all chromosomes)

Response: We agree that six SSR markers may be not enough to cover all chromosomes. However, the markers are independent and distributed along 4 chromosomes (V, VI, VII and X) out of the 10 basic chromosome number of this species. It should be noted that even though the markers AG7 and TC1 (on chromosome V) and AG8 and TC9 (on chromosome VII) are on the same chromosome, the genetic distance between them in a saturated genetic map of A. officinalis (Moreno et al., 2018) was high (> 120 cM), indicating that they are distributed in different regions of the asparagus genome.

In addition, those six SSR markers were very polymorphic in our populations; we detected a high number of alleles, and the PIC values were also high. On the other hand, the PCA analysis revealed different clusters and grouped the individuals by populations. Also, in a previous study (Garcia et al. 2021), these markers were also useful to assess the genetic variability of different genetic stocks. All taken together, these results indicate that the six markers could be enough to identify the genetic diversity present in our populations.

L 127… set of current cultivars… - how much cultivars, how much plants per cultivar???

Response: The information requested has been included in Material and Methods section.

                                                            (Ref. 23  4cvs?)

L 138 …two years… how old were the plants during first phenotyping?

Response: The information requested has been included in Material and Methods section.

L 143   date

 Response: It has been corrected

Results:

L 162  how much cultivars were used (n=63?) – 63 cultivars?

Response: The information requested has been included in Material and Methods section.

L 195   Tab.3 -  how you explain the increasing mean distance from the parental to „new diploid“ and

                          „Dipl. selected“?

Response: As we mentioned in the discussion, the greatest genetic variation detected in the new population compared to the parental population could be explained by the new genetic combinations that occurred after the crosses among plants of the origin population

L 230  Tab.5 – why no control (MH, cultivars) were evaluated ?

 Response: The controls were used only to compare their genetic variability with the new diploid population. Therefore, data of previous work, cited in the text, of these materials were used (Castro et al, 2013; Garcia et al, 2021). On the other hand, MH, the cultivars and the new diploid population were not growing in the same experimental plot.

Discussion:

-          See general comments

References:

Cited references are complete in text and reference list.

Reviewer 2 Report

For the discusiion section most part, the manuscript is well written. However, the 30% similarity excluding the reference is considered too high for a research article.  Nevertheless, the presentation of the manuscript did not convince me, and here I see some flaws that need to be rectified. 

Author Response

Comments and Suggestions for Authors

For the discusiion section most part, the manuscript is well written. However, the 30% similarity excluding the reference is considered too high for a research article.  Nevertheless, the presentation of the manuscript did not convince me, and here I see some flaws that need to be rectified. 

Response: Regarding the 30% similarity, it should be highlighted that most of the similarity is with publications that has been conducted by our research team (for example: sources 2, 4, 8, 11 and 13). Also, it is our understanding that the percentage of similarity is done by a software and, therefore, it does not consider the peculiarities of the present work. It should be noted that our work is the result of a long-term breeding program, and some results of the breeding program have been published in Garcia et al 2021, Castro et al 2014, Mousavizadeh et al 2018, Moreno et al 2006, Moreno et al 2008 (sources 2, 4, 8, 11 and 13 of the originality report), which are some of the papers with percentage of similarity. Most of the sentences with similarities are in the Material and Methods section. It should be noted that the techniques and methods used in the breeding program are usually the same but using different plant material. And that is the reason of the similarities. It is hard to be original when you are writing the Material and Methods section because you are describing standard procedures.  

Also, there are other word/sentences marked as similarities that, unfortunately, cannot be modified because there is no other way to write it. For instance: the name and address of our institution; the words ‘Introduction’ ‘introgressions’; the scientific names of asparagus species; ‘landrace Morado de Huetor’, among others.

So, we believe that if all the above mentioned is not considered as similarities, the percentage of similarity is lower than 33%.

Reviewer 3 Report

García et al. in their research article entitled ‘Genetic variability assessment of a diploid pre-breeding asparagus population developed using the tetraploid landrace ‘Morado de Huétor’generated a novel diploid plant population to enlarge the genetic base of asparagus using tetraploid introgressions as well as genetic diversity assessment of this population using SSR markers. The study provided a novel genetic resource and population developed approach in asparagus, which is of great significance for the genetic improvement in asparagus and may be of interest to researchers in this field. However, I have some minor concerns for this manuscript, the comments as follows:

Minor comments:
1. Six SSR markers were used for screening, please authors introduce introduce the development progress of SSR in this species, why only six markers were used, the author mentioned 12 markers were available in the method, whether these markers are polymorphic in the new population? In addition, please authors explain how to ensure the reliable and stable reproduction of genetic diversity using the limited amount of marker in population analysis.
2. The core work of this paper is to create a new population based on the authors' unique genetic material. It is suggested that the authors add the description of diploid and tetraploid characteristics, and provide evidence such as ploidy identification basis or flow DNA content measurement to support the reliability of the results in this paper.
3. In line 57-67, It is suggested that authors increased the introduction about the genetic population creation methods and the advantages of tetraploid population used for genetic improvement.

4. Please reformat Table 2 to make it easier to read.

Author Response

Comments and Suggestions for Authors

García et al. in their research article entitled ‘Genetic variability assessment of a diploid pre-breeding asparagus population developed using the tetraploid landrace ‘Morado de Huétor’’ generated a novel diploid plant population to enlarge the genetic base of asparagus using tetraploid introgressions as well as genetic diversity assessment of this population using SSR markers. The study provided a novel genetic resource and population developed approach in asparagus, which is of great significance for the genetic improvement in asparagus and may be of interest to researchers in this field. However, I have some minor concerns for this manuscript, the comments as follows:

Minor comments:
1. Six SSR markers were used for screening, please authors introduce introduce the development progress of SSR in this species, why only six markers were used, the author mentioned 12 markers were available in the method, whether these markers are polymorphic in the new population? In addition, please authors explain how to ensure the reliable and stable reproduction of genetic diversity using the limited amount of marker in population analysis.

Response: We used only six SSR markers because those markers were very polymorphic in our populations; we detected a high number of alleles, and the PIC values were also high. In addition, those six markers presented the most reliable and reproducible peak patterns among the 12 markers. On the other hand, the PCA analysis revealed different clusters and grouped the individuals by populations. Also, in a previous study (Garcia et al. 2021), these markers were also useful to assess the genetic variability of different genetic stocks. All taken together, these results indicate that the six markers could be enough to identify the genetic diversity present in our populations.

  1. The core work of this paper is to create a new population based on the authors' unique genetic material. It is suggested that the authors add the description of diploid and tetraploid characteristics, and provide evidence such as ploidy identification basis or flow DNA content measurement to support the reliability of the results in this paper.

Response: The characteristics of diploid cultivars and tetraploid landrace ‘Morado de Huetor ‘ are described in the Introduction (Lines 92-100).

We did not check the ploidy of the plants included in the new population (n=1000) in this work. The ploidy level of the parental population (n=77) was assessed in a previous study (Castro et al 2014). The new population was generated from the diploid collection described in Castro et al 2014, which is the parental population in the present study. In Castro et al 2014 we discussed the following:

It is assumed that gametes carrying an extra-long chromosome are genetically more unbalanced than gametes with a shorter chromosome and results in increased gametic and zygotic lethality (Singh, 2010). The transmission of extra chromosomes in plants is considered different in female and in pollen. Theoretically, about 50% of the trisomic plants are expected in the progeny from2x + 1 × 2x, but this percentage is rarely observed (Singh, 2010). The transmission of extra chromosomes through pollen is usually very low. Pollen with n + 1 chromosome constitution is unbalanced and generally unable to compete in fertilization with pollen carrying the balanced, n, chromosome number (Gupta, 2007; Singh, 2010). According to Löptien (1979), in asparagus, the transmission rates of an extra chromosome seems to be higher for the eggs than for pollen and the larger chromosomes have lower pollen transmission than the smaller ones. In a practical point of view, collecting seeds from female plants of the diploid population would favor the disappearance of the trisomic plants after several generations, resulting in a completely diploid population. In spite of the lack of accuracy of flow cytometry to distinguish between diploid and trisomic asparagus in this study, this method is more user-friendly and faster than microscope observations.”  

  1. In line 57-67, It is suggested that authors increased the introduction about the genetic population creation methods and the advantages of tetraploid population used for genetic improvement.

Response: The information requested has been included in the Introduction section. The advantages of tetraploid population used for genetic improvement are detailed in the manuscript (lines 92-100)

  1. Please reformat Table 2 to make it easier to read.

Response: We agree with the reviewer 3 regarding to the table 2.  We have decided to remove this table because it is not relevant for the results and most information described in Table 2 is also described in the text of the manuscript.

Round 2

Reviewer 2 Report

The authors' reasons for high similarity were justified 

Author Response

Dear reviewer,

Thank you for your positive comment regarding to our justification of the similarity rate of our manuscript

The authors